# Automated Early-Stage Glaucoma Detection Using a Robust Concatenated AI Model

**DOI:** 10.3390/bioengineering12050516

**Published:** 2025-05-13

**Authors:** Wheyming Song, Ing-Chou Lai

**Affiliations:** 1Department of Information Engineering and Computer Science, Feng Chia University, Taichung 407102, Taiwan; 2Department of Ophthalmology, Chiayi Chang Gung Memorial Hospital, Puzi City 61363, Taiwan; lai1@cgmh.org.tw

**Keywords:** glaucoma, irreversible blindness, peripapillary retinal nerve fiber layer thickness, macular thickness

## Abstract

Glaucoma is a leading cause of irreversible blindness worldwide; therefore, detection of this disease in its early stage is crucial. However, previous efforts to identify early-stage glaucoma have faced challenges, including insufficient accuracy, sensitivity, and specificity. This study presents a concatenated artificial intelligence model that combines two types of input features: fundus images and quantitative retinal thickness parameters derived from macular and peri-papillary retinal nerve fiber layer (RNFL) thickness measurements. These features undergo an intelligent transformation, referred to as “smart preprocessing”, to enhance their utility. The model employs two classification approaches: a convolutional neural network approach for processing image features and an artificial neural network approach for analyzing quantitative retinal thickness parameters. To maximize performance, hyperparameters were fine-tuned using a robust methodology for the design of experiments. The proposed AI model demonstrated outstanding performance in early-stage glaucoma detection, outperforming existing models; its accuracy, sensitivity, specificity, precision, and $F_{1}$-Score all exceeding 0.90.

## 1. Introduction

Glaucoma is a chronic eye disease that gradually damages the optic nerve and impairs vision, leading to a decreased quality of life. Unfortunately, glaucoma is an irreversible eye disease, and while medical or surgical interventions can slow its progression, they cannot restore eye health. According to Allison et al. [1], approximately 76 million people worldwide have been diagnosed with glaucoma.

The primary imaging techniques for glaucoma detection are optical coherence tomography (OCT) and ocular fundus camera (OFC) imaging. These techniques provide critical information such as fundus images, macular thickness, and peripapillary nerve fiber layer thickness. The Humphrey Field Analyzer (HFA) device measures the vision mean defect (V-MD) values to classify glaucoma severity as mild (also called early), moderate, or advanced. Specifically, the values can be interpreted as follows:V-MD values between 0 dB and −6 dB indicate mild (early) glaucoma;V-MD values between −6 dB and −12 dB indicate moderate glaucoma;V-MD values below −12 dB indicate advanced glaucoma.

Detecting early-stage glaucoma is essential because the damage it causes is irreversible.

Previous studies on glaucoma detection have not resulted in a well-established means of detecting early-stage glaucoma. Most studies before 2017 (including [2,3,4,5,6,7,8,9]) used quantitative structural features as the primary input for glaucoma detection. Since 2018, research (including [9,10,11,12,13,14,15,16,17,18,19,20,21]) has increasingly utilized color fundus images; however, only a limited number of studies have specifically addressed early-stage glaucoma detection. Among these studies, the highest reported accuracy [17] was 0.85. However, Ref. [17] did not provide details on the sensitivity, specificity, precision, or $F_{1}$-Score of the method. Furthermore, none of the studies appears to have utilized a statistically robust design of experiments (DOE) (see Section 2.1.3) to optimize its hyperparameters.

This paper focuses on detecting early-stage glaucoma and seeks to address the shortcomings of previous studies. Section 2 outlines the proposed approach, which integrates two types of input features (images and retinal thickness metrics) through their corresponding classification models. These features are enhanced using a “smart pre-processing” technique to improve their effectiveness. The model’s hyperparameters are optimized using the design of experiments (DOE) approach. Section 3 presents the results, evaluating the performance of the proposed model. Finally, Section 4 provides a discussion that summarizes the key findings, insights, and limitations of the study.

## 2. Materials and Methods

The framework of the proposed method is illustrated in Figure 1, in which there are four main parts: (A) data collection and design of experiments, (B) feature input, (C) smart pre-processing, and (D) classification. Further discussion of the abovementioned four parts can be found below.

### 2.1. Data Collection and Design of Experiments

The proposed framework utilizes a dataset of 416 samples (204 healthy individuals and 212 individuals with glaucoma: 130 early, 51 moderate, and 31 severe) provided by Kaohsiung Chang Gung Memorial Hospital in Taiwan. The dataset was divided into training, validation, and test sets, with proportions of approximately 50%, 20%, and 30%, respectively. These proportions were consistently applied across all categories, including healthy, early, moderate, and severe cases. For example, 50%, 20%, and 30% of the early-stage glaucoma cases were allocated to the training, validation, and test sets, respectively. The design of experiments involved 5 blocks, with each block consisting of 200 epochs and 100 batches per epoch.

#### 2.1.1. Blocks and Epochs

First, we will discuss the relationship between blocks and epochs. The proposed framework contains *l* blocks, each with $h_{i}$ epochs, where l=5 and $h_{i}\leq h,i=1,2,\ldots ,\;l$, h=200. If the termination rule was not reached, then $h_{i}=h$. After the final epoch was completed for each block, we constructed a so-called block-model to apply to the test sets.

#### 2.1.2. Epochs and Batches

Next, we will discuss the relationship between epochs and batches. Each epoch passed through b=100 batches, each with batch size (images) m=40. The m=40 images in each batch were randomly selected from the training set in the associated block. Each image in each batch passed through the pre-CNN process (Steps 1 to 5, as shown in Figure 4) and the proposed concatenated model (as shown in Figure 5). At the end of each batch, the loss function and therefore the associated parameters were updated. A pre-validation model was derived from 40 batches of 40 images.

#### 2.1.3. Robust DOE

Here, we will explain the process of identifying the optimal hyperparameter values for the adopted concatenated model. The initial design of experiments (DOE) involved nine factors: five factors for Model 1 (CNN), two factors for Model 2 (ANN), and two factors for the final concatenated model. These factors are specified as follows.

There were five factors for Model 1 (CNN), which uses fundus images as input data:$A_{1}$: image size used;$B_{1}$: convolutional layers;$C_{1}$: number of kernels in each convolutional layer;$D_{1}$: treatment for the area outside the ROI;$E_{1}$: parameters α and β used in the proposed loss function (see Figure 2) for the training data:(1)$$L=-y{\left(1-\hat{p}\right)}^{\alpha }\log\;{\hat{p}}^{\beta }-\left(1-y\right){\left(\hat{p}\right)}^{\alpha }\log{\left(1-\hat{p}\right)}^{\beta },$$
where the terms are defined as follows:-The response y=1 or y=0 indicates an early-stage glaucoma sample or a normal sample, respectively.-$\hat{p}$, a sigmoid function, gives the estimate of the true value *y*. Regarding the test set, the estimate of *y* for the test set data is $\hat{y}=1$ if $\hat{p}>0.5$; otherwise, the estimate is $\hat{y}=0$.

There were two factors for Model 2 (ANN), which uses transformed macular thickness and peripapillary retinal nerve fiber layer thickness as input data:Factor $A_{2}$: No. of layers in the ANN;Factor $B_{2}$: No. of nodes in the ANN.

There were two factors for the final concatenated model (referred to as Model 3):Factor $F_{1}$: No. of layers in the concatenated model;Factor $F_{2}$: No. of nodes in the concatenated model.

Based on the results of [18], authored by the same first author as this paper, we decided to use the same optimal values for the first five factors ($A_{1}$–$E_{1}$) and create a new DOE for the remaining four factors ($A_{2}$, $B_{2}$, $A_{3}$, $B_{3}$). The optimal values for the first five factors are outlined in Section 2.4.1. For the last four factors, a new DOE was created, with the factors redefined and grouped as follows:Factor $A_{3}$ (formerly $A_{2}$, number of layers in Model 2) with two levels: −1 (2 layers), 1 (4 layers)Factor $B_{3}$ (formerly $B_{2}$, number of layers in Model 2) with two levels: −1 (4 nodes), 1 (16 nodes)Factor $C_{3}$ (formerly $F_{1}$, number of layers in Model 3) with two levels: −1 (2 layers), 1 (3 layers)Factor $D_{3}$ (formerly $F_{2}$, number of layers in Model 3) with two levels: −1 (4 nodes), 1 (16 nodes)

A full $2^{4}$ factorial design was implemented for these factors as follows. We first adopted a $2^{4}$ full factorial design for Model 2, where there were four factors (denoted as $A_{3},B_{3},C_{3}$, and $D_{3}$). The results of the 16 combinations (runs) of the $2^{4}$ full factorial design are listed in Table 1, where the mean accuracy and the standard error are shown in the last two associated columns. The listed data in the rightmost three columns are explained as follows: ${\underline{g}}_{j}^{\left(i\right)}$ denotes a vector containing $g_{j}^{\left(i\right)}\left(k\right)$, the accuracy at the *i*th replication, *j*th combination, and *k*th block, where i=1,2;j=1,2,…,8;k=1,2,…,10. The associated mean and standard deviation are denoted as ${\bar{g}}_{j}^{\left(i\right)}$ and $s_{j}^{\left(i\right)}$. Finally, the optimal values are described in Section 2.4.2 and Section 2.4.3.

### 2.2. Input Features

The two types of input features were (1) fundus images and (2) quantitative retinal thickness parameters, which are functions of macular thickness and retinal nerve fiber layer (RNFL) thickness. The fundus images, macular, and RNFL are highlighted in blue and shown in Figure 3 as parts (a), (b), and (c), respectively.

#### Fundus Image Feature

Fundus images are photographs of the interior surface of the eye, specifically the retina, which is the light-sensitive layer at the back of the eye. These images are typically captured using a specialized device called a fundus camera. A fundus image shows key anatomical structures of the eye, including the following:Retina: the light-sensitive tissue that detects visual information and sends signals to the brain;Optic disc: the region where the optic nerve exits the eye, appearing as a bright circular area;Macula: the central part of the retina responsible for detailed vision and color vision;Blood vessels: the network of veins and arteries that supply blood to the retina.

In this research, fundus images serve as one type of input data. Although fundus images include structures such as the retina and macula, we use the images solely as inputs. Instead of fundus images, the information related to the retina and macula is represented by their associated thickness.

### 2.3. Smart Pre-Processing

#### 2.3.1. Fundus Images

Here, we first explain the pre-processing for input feature 1, (fundus image), which includes Input Images (Step 1), Clean Images (Step 2), Retinex (Step 3), Region of Interest (Step 4), and Dimension Changes (Step 5). The abovementioned 5 steps are illustrated in Figure 4 and are shortly explained below. A clear explanation can also be found in the literature [19].

Step 1. Input Images: All input fundus images include text about patient information and the date that the associated images were collected.Step 2. Clean Images: All text on each image was replaced by pixels with a value of 0, which is the color black.Step 3. Retinex: The word “Retinex” combines “retina” and “cortex”, where the retina is the part of the eye that detects color, while the visual cortex is the part of a brain that processes the information it receives from the retina. This step was used to remove the light effect from flash images.Step 4. Region of Interest (ROI*):Based on our prior knowledge, the region of interest should be near the optic nerve cup, and macular portion.Accordingly, we selected the (ROI*) as follows.(2)$$I_{\mathrm{ROI}}^{\left(i\right)}\left(x,y\right)=\left\{\begin{array}{cc} R_{III}^{\left(i\right)}\left(x,y\right), & x\;\epsilon \;{\mathrm{ROI}}^{*}\\ 0, & \mathrm{otherwise}, \end{array}\right.$$
where ROI* covers the optic nerve cup, optic disc, and macula.Step 5. Image augmentation and dimension reduction:Each input image was randomly rotated 30, 60, 90, 120, or 180 counterclockwise. Thus, each image was augmented into two images. Finally, we reduced each image, shifting the row and column sizes of 500×750 into smaller sizes 78×116. Let D(i) denote the image processed at the end of this step, where i=1,2,3 regarding RGB.

**Figure 4 bioengineering-12-00516-f004:**
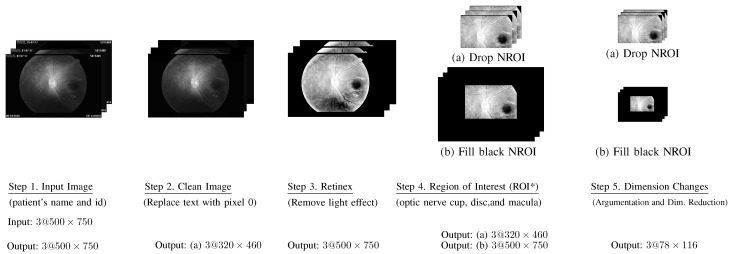
Pre-CNN process (steps 1–5). Then connect to (I) in Figure 5.

#### 2.3.2. Quantitative Measure

This subsection explains the pre-processing for input Feature 2: the quantitative structural feature, which is a function of macular thickness and retinal nerve fiber layer thickness.

Regarding to macular thickness, we adopted four parts:UO: denotes Upper Outer;UI: denotes Upper Inner;LO: denotes Lower Outer’LI: denotes Lower Inner.

The associated areas of UO, UI, LO, and LI are 1.6875π, 0.5 π, 0.5 π, and 1.6875π, where $1.6875=[3^{2}-{\left(1.5\right)}^{2}]/4$ and $0.5=[{\left(2.5\right)}^{2}-{\left(0.5\right)}^{2}]/4$; see Figure 6. The radii $\bar{OA},\bar{OB},\bar{OC}$ of the three circles are 0.5, 1.5, and 3, respectively.

The weighted average of MT (named WA_MT) is defined below.(3)$$\mathrm{WA}\text{\_}\mathrm{MT}:\frac{0.5(\mathrm{UI}+\mathrm{LI})+1.6875(\mathrm{UO}+\mathrm{LO})}{2(0.5+1.6875)},$$

Regarding the retinal nerve fiber layer (RNFL) thickness, we adopted seven parts:G: denotes Global, whose area is 1;T: denotes Temporal, which has area 4;TS: denotes Temporal Superior, which has area 2;TI: denotes Temporal Inferior, which has area 2;N: denotes Nasal, which has area 4;NS: denotes Nasal Superior, which has area 2;NI: denotes Nasal Inferior, which has area 2.

Based on the area for part 1 (G) is 1, we provide an estimate area for the rest of 6 parts, as listed above. The total area is 17. The ratio among the 7 parts for calculating RNFL thickness is illustrated in Figure 7.

The weighted average of RNFL (denoted as WA_RNFL) is defined as follows:(4)$$\mathrm{WA}\text{\_}\mathrm{RNFL}=\frac{\left(4/17)(T+N)+(2/17)(\mathrm{TS}+\mathrm{NS}+\mathrm{TI}+\mathrm{NI}\right)}{2(8/17)}$$

### 2.4. Classification

The proposed AI models corresponding to two types of input features and the final concatenated model are discussed in Section 2.4.1, Section 2.4.2 and Section 2.4.3.

#### 2.4.1. Pre-Concatenated Model 1

The input feature of the Pre-Concatenated model 1 is fundus image (Input Feature 1). Pre-Concatenated model 1 can be views as the traditional CNN model deleting all parts after flatten. We use notation a@b×b to indicate that there are a images, each with dimension b×c. For example, 32@78×116 indicates that there are 32 images, each with dimension 78×116.

Pre-Concatenated Model 1 contains 5 steps, in which the first 4 steps contains Convolution, Normalization, ReLU, and Pooling (C-N-R-P).

Step 5.1. C-N-R-P(1):C: 32@78×116;N: 32@78×116;R: 32@78×116;P: 32@39×58.

Step 5.2. C-N-R-P(2):C: 32@18×28;N: 32@18×28;R: 32@18×28;P: 32@16×26.

Step 5.3. C-N-R-P(3):C: 32@8×13;N: 32@8×13;R: 32@8×13;P: 32@6×11.

Step 5.4. C-N-R-P(3):C: 32@6×11;N: 32@6×11;R: 32@6×11;P: 32@3×5.

Step 5.5. C: one convolution with kernel matrix 3×5, which yields one vector with 32 values.

The adopted hyperparameters for Pre-Concatenated Model 1 are as follows:(1)$A_{1}$: sizes used in images: 78×116;(2)$B_{1}$: convolutional layers: 4×4=16;(3)$C_{1}$: number of kernels in each convolutional layer: 32 (which yields 32 images);(4)$D_{1}$: treatment for the area outside the ROI: delete (rather than fill with black);(5)$E_{1}$: parameter β used in the proposed loss function: parameter α=1, β=1.5 used in the loss function defined in Equation (Equation 1).

The optimal values for the listed hyperparameters were determined through a design of experiments (DOE) approach, as described in [18].

#### 2.4.2. Pre-Concatenated 2: ANN

Pre-Concatenated 2 model is an artificial neural network (ANN) that uses WA_MT and WA_RNFL as its two input values. Recall that WA_MT and WA_RNFL are derived from macular thickness and RNFL thickness, respectively.

Factor $A_{2}$ (No. of layers in ANN): 4 layers;Factor $B_{2}$ (No. of nodes in ANN): 16 nodes.

We will now explain how we ultimately arrived at proposing Model 2, which uses WA_MT and WA_RNFL as its two input values.

A comprehensive experiment was conducted to evaluate various combinations of 17 potential input values, grouped as follows:macular thickness(1)UO: Upper Outer(2)UI: Upper Inner;(3)LO: Lower Outer;(4)LI: Lower Inner;(5)Weighted average for the Upper region: average of UO and UI;(6)Weighted average for the Lower region: average of LO and LI;(7)Weighted average for the Inner region: average of UI and LI;(8)Weighted average for the Outer region: average of UO and LO;(9)Overall weighted average of macular thickness: WA_MT, as defined in Equation (Equation 3).RNFL thickness(1)G: Global;(2)T: Temporal;(3)TS: Temporal Superior;(4)TI: Temporal Inferior;(5)N: Nasal;(6)NS: Nasal Superior;(7)NI: Nasal Inferior;(8)Overall weighted average of RNFL thickness: WA_RNFL, as defined in Equation (Equation 4).

After thoroughly analyzing the results, we determined that WA_MT and WA_RNFL are the most effective input values for Model 2. To save space, the detailed results for various combinations of inputs are omitted here.

#### 2.4.3. Proposed AI Model 3: Concatenated Model

The final proposed AI model is a concatenated model that combines Model 1 and 2.

–Factor $A_{3}$ (No. of layers in Concatenate): 2 layers;–Factor $B_{3}$ (No. of nodes in Concatenate): 16 nodes.

## 3. Results

### 3.1. Commonly Used Performance Metrics in Glaucoma Detecting

Before defining commonly used performance metrics for glaucoma detection, we first explain the concepts of the actual condition (also referred to as the “fact” as shown in Figure 8) (whether someone has glaucoma or not) and the test results (positive or negative). Here, “positive” refers to the test result indicating that the person has glaucoma, while “negative” refers to the test result indicating that the person does not have glaucoma. It is important to note that the terms “positive” and “negative” should not be used to describe the actual condition (fact).

Next, we define true positives (TP), true negatives (TN), false positives (FP), and false negatives (FN) as follows, as illustrated in Figure 8.

–TP: correctly identifying a positive test result, where “positive” refers to the result of the test rather than the factual condition;–TN: correctly detecting a negative case;–FP: incorrectly detecting a negative case as positive;–FN: incorrectly detecting a positive case as negative.

Now, we define the commonly used performance metrics for glaucoma detection, which include accuracy, sensitivity (also known as recall), specificity, precision, the $F_{\beta }$-score as follows:(1)Accuracy: (TP+TN)/(TP+FP+FN+TN).(2)Sensitivity, also referred to as recall: TP/(TP+FN),shown in Figure 8a,b. The insight of using recall is reflected in the name itself: it reflects how much the model recalls of the TP (true positive) results.(3)Specificity: TN/(FP+TN), shown in Figure 8a.(4)Precision: TP/(TP+FP), shown in Figure 8b.(5)$F_{\beta }$-Score:$$\frac{\left(1+\beta^{2}\right)\left(\mathrm{Precision}\times \mathrm{Recall}\right)}{\left(\beta^{2}\mathrm{Precision}+\mathrm{Recall}\right)}.$$(6)$F_{1}$-Score (also referred to as the letter H):$$\frac{2(\mathrm{Precision}\times \mathrm{Recall})}{\mathrm{Precision}+\mathrm{Recall}}.$$The F1-Score is a special case of the $F_{\beta }$-Score with β=1, which is referred to as the letter H because it is the harmonic mean of recall and precision, as shown in Figure 9.

These metrics are crucial for evaluating the performance of glaucoma detection systems. It is important to note that the common feature of sensitivity and specificity is that their numerators are both related to the actual condition (fact), while the common feature of recall and precision is that their numerators are both related to true positive (TP) results.

To ensure completeness, in addition to H, we also include the arithmetic mean (denoted as A) and the geometric mean (denoted as G), as follows.

–A: The letter A represents the average mean of recall and precision, which is calculated as (precision+recall)/2.–G: The letter G is referred to as the geometric mean of recall and precision, which is calculated as $\sqrt{\mathrm{precision}\times \mathrm{recall}}$.

The reason for using H instead of A or G as the performance measure for glaucoma detection is that H reflects a value closer to min(precision,recall). Note that H≤G≤A. Therefore, H (instead of A or G) is chosen as the performance measure.

### 3.2. Determine the Optimal Values of the Four Factors Used in the Concatenated Model

The results shown in Table 2 indicate that the coefficients for all factors, except Factor C, are positive. Consequently, the optimal values for the four factors in the Concatenated Model are as follows:–Factor $A_{3}$ (No. of layers in ANN): four layers;–Factor $B_{3}$ (No. of nodes in ANN): sixteen nodes;–Factor $C_{3}$ (No. of layers in Concatenate): two layers;–Factor $D_{3}$ (No. of nodes in Concatenate): sixteen nodes.

These choices are supported by an analysis of the main effects and interaction effects, as illustrated in Figure 10 and Figure 11, respectively.

The top five combinations of Factors $A_{3}$ through $D_{3}$ are summarized in Table 3.

### 3.3. Performance of the Concatenated Model

The performance of the model in detecting early-stage glaucoma is summarized in Table 4, including the estimated accuracy, sensitivity, specificity, precision, and $F_{1}$-Score, along with their associated standard errors. For Model 1, all performance metrics are below 0.9, while for Model 2, they are below 0.7. In contrast, the concatenated model achieves superior performance, with all metrics exceeding 0.9, including accuracy of 0.93, sensitivity of 0.91, specificity of 0.94, precision of 0.92, and an $F_{1}$-Score of 0.90.

Model 2 performs worse than Model 1, possibly due to its limited input values (WA_MT and WA_RNFL). However, the concatenated model outperforms Model 2 after incorporating that mode, indicating that Model 2 makes a significant contribution.

## 4. Discussion

This paper introduces an efficient artificial intelligence (AI) model for the automatic detection of early-stage glaucoma. The proposed framework utilizes a dataset of 416 data points (204 healthy cases and 212 cases of glaucoma, divided into 130 early, 51 moderate, and 31 severe) from Kaohsiung Chang Gung (KCG) Memorial Hospital in Taiwan. The AI model integrates two types of input features (image and structural features) along with two corresponding classification models. The image feature is based on fundus images, while the structural feature is derived from macular thickness and retinal nerve fiber layer thickness. The two classification models used are a convolutional neural network (CNN) for the image feature and an artificial neural network (ANN) for the structural feature. Furthermore, a design of experiments (DOE) approach is employed to optimize the hyperparameters of the proposed model.

In our previous work ([18]), referenced in the literature review, we provided an indepth comparison with existing state-of-the-art methods, showing that the proposed CNN model outperforms these techniques. As a result, a detailed re-evaluation is not necessary in this study.

The key contributions of this paper can be summarized as follows:–Effectiveness: The proposed model achieves an accuracy of 0.93 for early-stage glaucoma detection, which is significantly higher than the 0.85 accuracy reported in [17]. Additionally, Ref. [17] did not report metrics such as sensitivity, specificity, precision, or $F_{1}$-Score. In contrast, the concatenated model demonstrates superior performance, with all metrics exceeding 0.9: accuracy of 0.93, sensitivity of 0.91, specificity of 0.94, precision of 0.92, and $F_{1}$-Score of 0.90.–Robustness: The standard errors for the estimated performance are all below 0.04. To the best of our knowledge, none of the previous studies appears to have used a statistically robust design of experiments (DOE) to optimize hyperparameters.–Expert Knowledge Integration: For any problem, such as detecting glaucoma, experts can always provide valuable insights. For example, they know that the thickness of the macular or peripapillary retinal nerve fiber layer is smaller in glaucoma cases than in normal cases. We believe that a concatenated model that incorporates expert knowledge should outperform one that relies solely on image data. Specifically, research from journal papers published after 2018 (including [9,10,11,12,13,14,15,16,17,18,19,20,21]) suggests that incorporating expert knowledge will certainly lead to better performance compared to using image data alone.–practicality for clinical use: The model requires approximately 1 min to process a single block (blocks are discussed in Section 2.1), demonstrating its practicality for clinical use. The computational system used in our study consisted of the following hardware:(1)Motherboard: ASUS, ROG STRIX Z590-E GAMING WIFI;(2)CPU: 11th Gen Intel(R) Core(TM) i9-11900KF @ 3.50 GHz/eight cores;(3)GPU: NVIDIA [GeForce RTX 3090], NVIDIA [GeForce RTX 2080 Ti];(4)Graphics card: NVIDIA [GeForce RTX 3090], NVIDIA [GeForce RTX 2080 Ti];(5)RAM: 64 GB DDR4 (16 GB X 4);(6)Operating system: Debian Linux 12;(7)Others: HDDs: 1 TB X 1, 6 TB X 4.

Although the model demonstrates promising performance, the potential risks associated with AI-based diagnosis, such as excessive reliance on automated systems or the misinterpretation of results, must be carefully evaluated in clinical environments. It is crucial that the model serves as a supportive tool for healthcare professionals, rather than replacing clinical judgment.

We also acknowledge that using data from a single hospital in Taiwan may limit the model’s generalizability, as glaucoma presentation can vary across ethnic groups due to genetic and environmental factors. While external validation would enhance robustness, this study primarily focuses on developing and evaluating the model within the available dataset. Future work should focus on multi-center validation with diverse populations to assess the model’s robustness across ethnic and demographic variations. Despite this limitation, the core contribution of this study is the proposed AI framework, which remains adaptable across different datasets, even if specific parameters require adjustment.

## Figures and Tables

**Figure 1 bioengineering-12-00516-f001:**
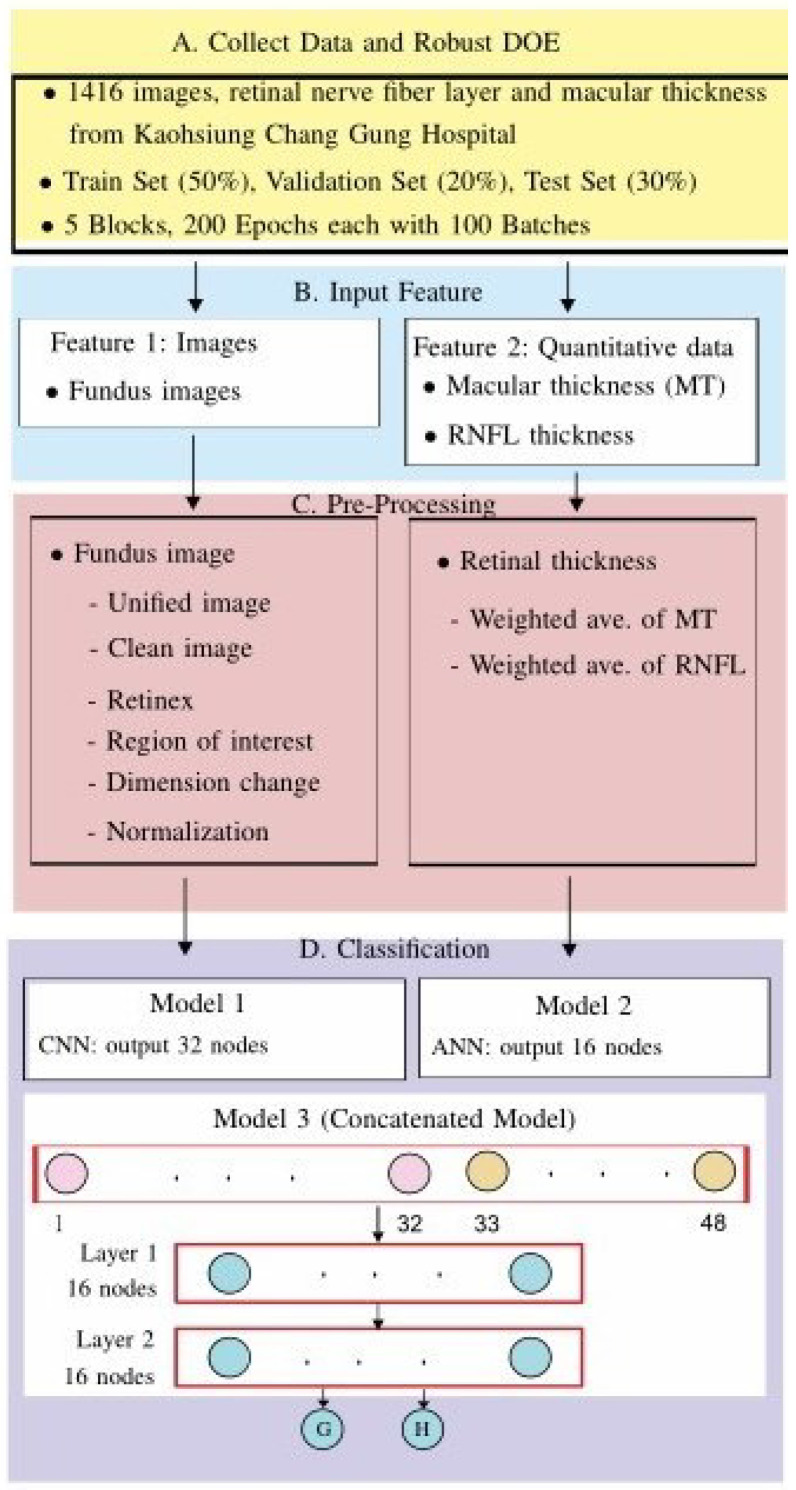
Proposed model.

**Figure 2 bioengineering-12-00516-f002:**
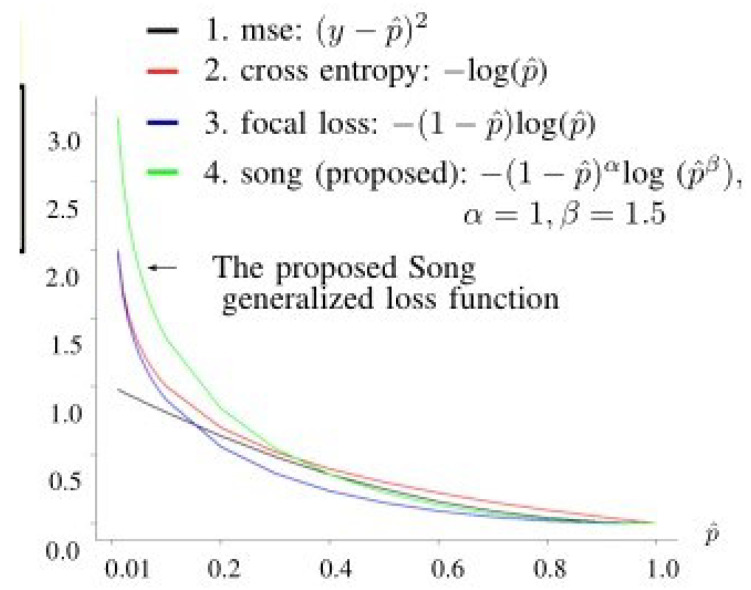
Four loss functions, assuming y=1.

**Figure 3 bioengineering-12-00516-f003:**
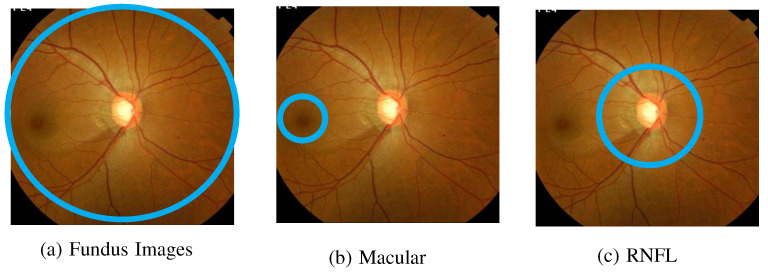
Input features for a right eye.

**Figure 5 bioengineering-12-00516-f005:**
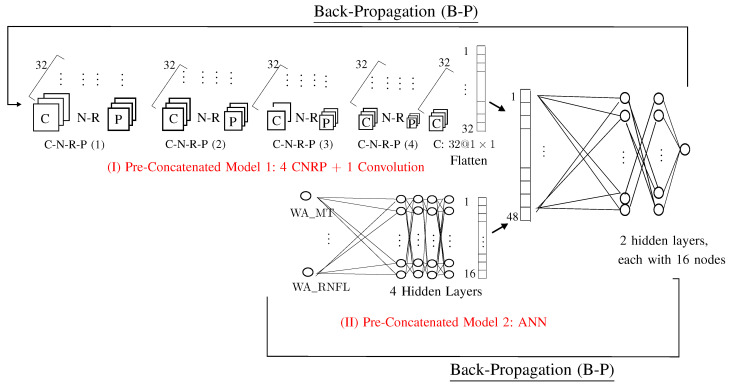
Proposed concatenated model.

**Figure 6 bioengineering-12-00516-f006:**
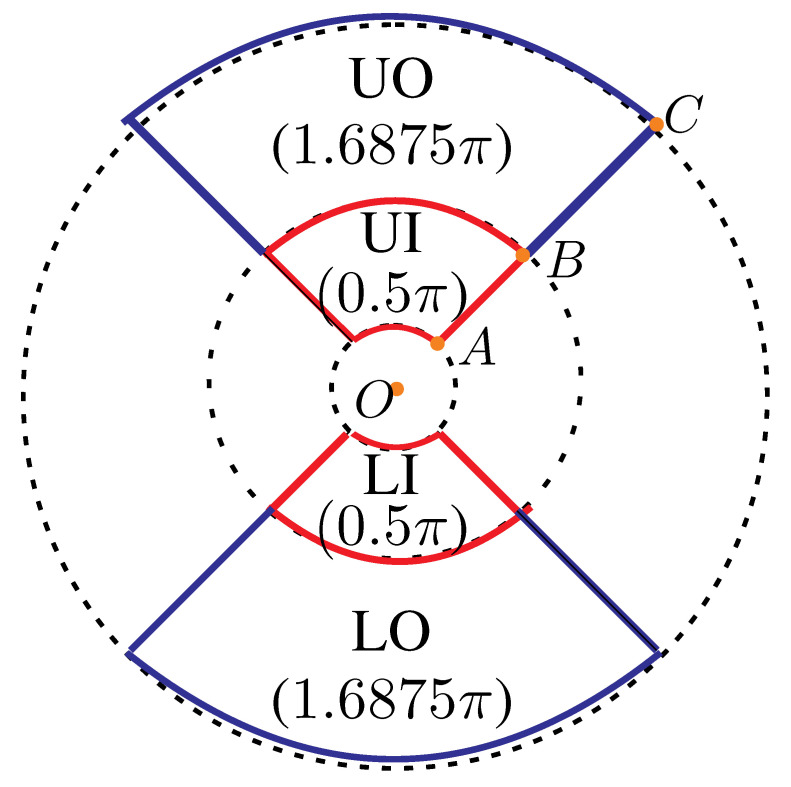
Weighted average of macular thickness.

**Figure 7 bioengineering-12-00516-f007:**
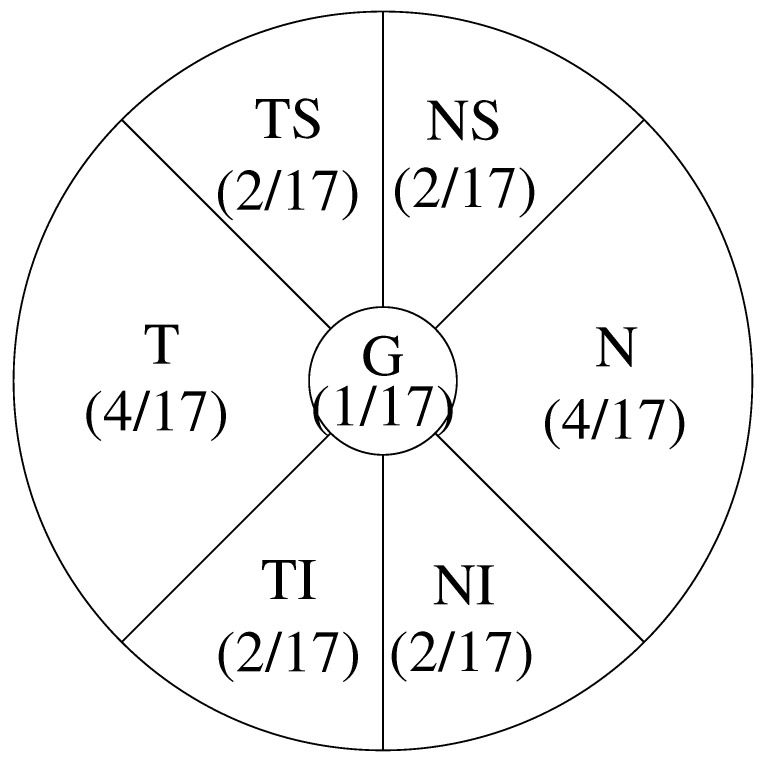
Weighted average of RNFL thickness.

**Figure 8 bioengineering-12-00516-f008:**
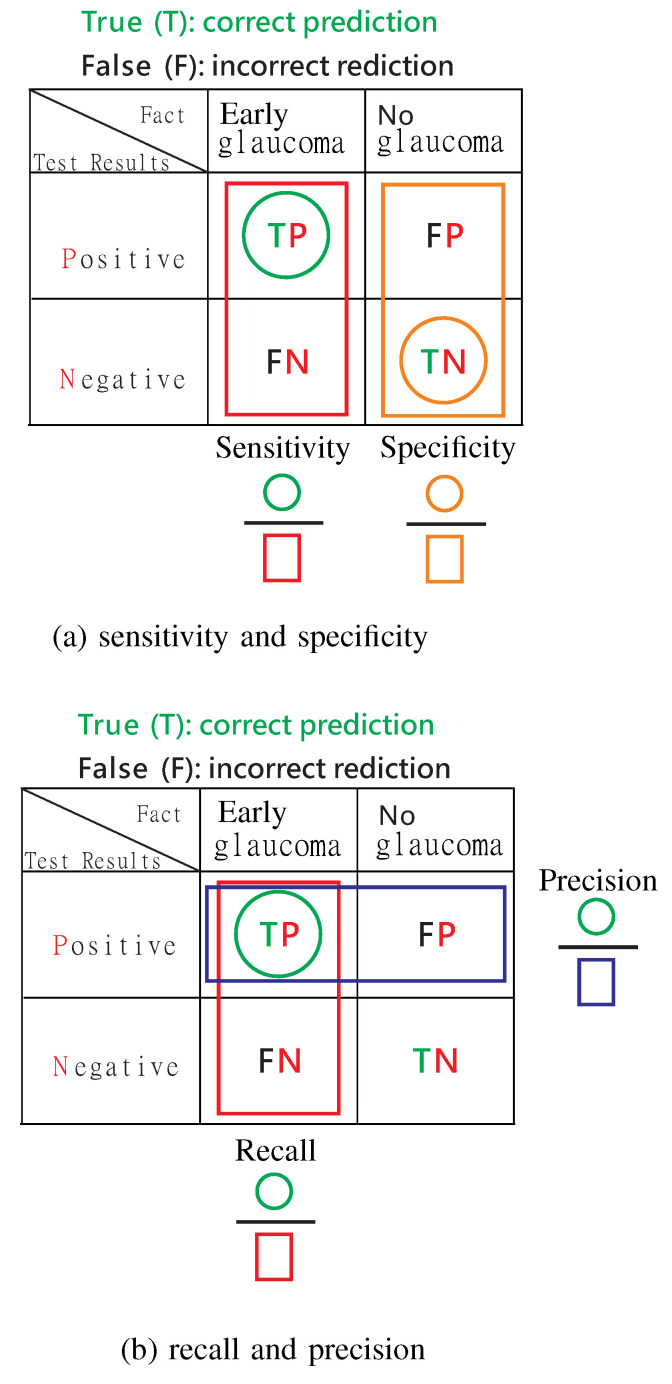
Performance measures.

**Figure 9 bioengineering-12-00516-f009:**
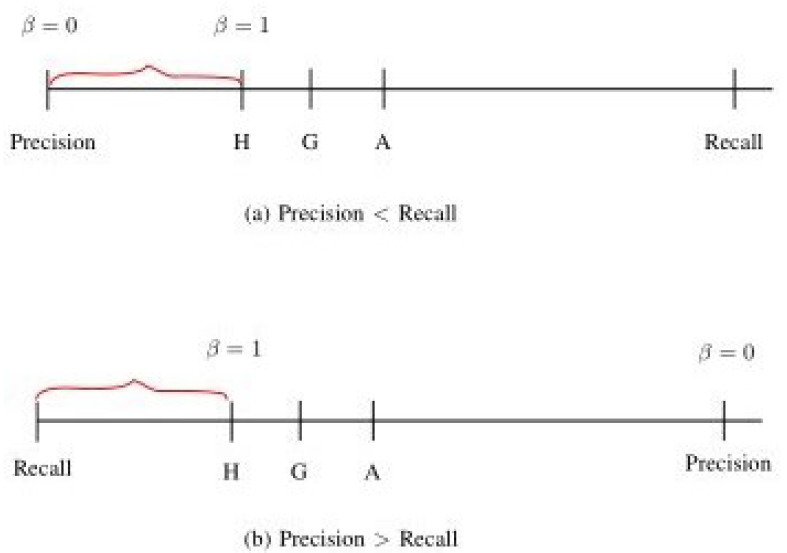
Performance measures: $F_{\beta }$-Score.

**Figure 10 bioengineering-12-00516-f010:**
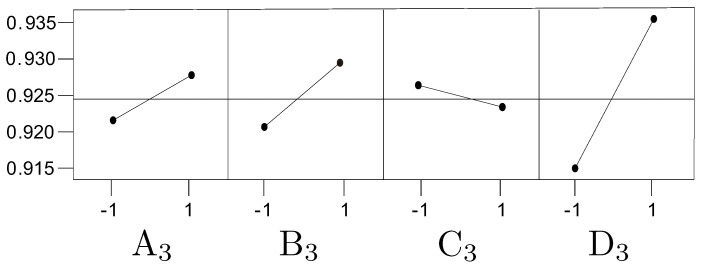
AI model 3, main effect: accuracy.

**Figure 11 bioengineering-12-00516-f011:**
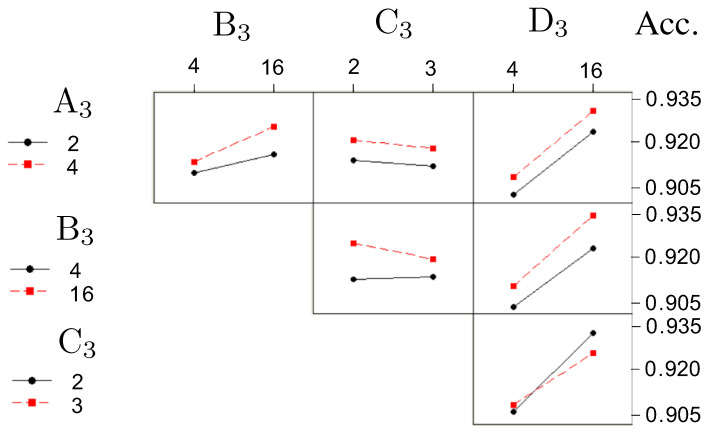
AI model 3, interaction effect: accuracy.

**Table 1 bioengineering-12-00516-t001:** The $2^{4}$ full factorial design, with two replicates.

Comb. (Run)	$A_{3}$, $B_{3}$, $C_{3}$, $D_{3}$	Ori. Data	Mean Accuracy	Stand. Error
1	− − − −	${\underline{g}}_{1}^{\left(1\right)},\;{\underline{g}}_{1}^{\left(2\right)}$	${\bar{g}}_{1}^{\left(1\right)},\;{\bar{g}}_{1}^{\left(2\right)}$	$s_{1}^{\left(1\right)},\;s_{1}^{\left(2\right)}$
2	+ − − −	${\underline{g}}_{2}^{\left(1\right)},\;{\underline{g}}_{2}^{\left(2\right)}$	${\bar{g}}_{2}^{\left(1\right)},\;{\bar{g}}_{2}^{\left(2\right)}$	$s_{2}^{\left(1\right)},\;s_{2}^{\left(2\right)}$
3	− + − −	${\underline{g}}_{3}^{\left(1\right)},\;{\underline{g}}_{3}^{\left(2\right)}$	${\bar{g}}_{3}^{\left(1\right)},\;{\bar{g}}_{3}^{\left(2\right)}$	$s_{3}^{\left(1\right)},\;s_{3}^{\left(2\right)}$
4	+ + − −	${\underline{g}}_{4}^{\left(1\right)},\;{\underline{g}}_{4}^{\left(2\right)}$	${\bar{g}}_{4}^{\left(1\right)},\;{\bar{g}}_{4}^{\left(2\right)}$	$s_{4}^{\left(1\right)},\;s_{4}^{\left(2\right)}$
5	− − + −	${\underline{g}}_{5}^{\left(1\right)},\;{\underline{g}}_{5}^{\left(2\right)}$	${\bar{g}}_{5}^{\left(1\right)},\;{\bar{g}}_{5}^{\left(2\right)}$	$s_{5}^{\left(1\right)},\;s_{5}^{\left(2\right)}$
6	+ − + −	${\underline{g}}_{6}^{\left(1\right)},\;{\underline{g}}_{6}^{\left(2\right)}$	${\bar{g}}_{6}^{\left(1\right)},\;{\bar{g}}_{6}^{\left(2\right)}$	$s_{6}^{\left(1\right)},\;s_{6}^{\left(2\right)}$
7	− + + −	${\underline{g}}_{7}^{\left(1\right)},\;{\underline{g}}_{7}^{\left(2\right)}$	${\bar{g}}_{7}^{\left(1\right)},\;{\bar{g}}_{7}^{\left(2\right)}$	$s_{7}^{\left(1\right)},\;s_{7}^{\left(2\right)}$
8	+ + + −	${\underline{g}}_{8}^{\left(1\right)},\;{\underline{g}}_{8}^{\left(2\right)}$	${\bar{g}}_{8}^{\left(1\right)},\;{\bar{g}}_{8}^{\left(2\right)}$	$s_{8}^{\left(1\right)},\;s_{8}^{\left(2\right)}$
9	− − − +	${\underline{g}}_{9}^{\left(1\right)},\;{\underline{g}}_{9}^{\left(2\right)}$	${\bar{g}}_{9}^{\left(1\right)},\;{\bar{g}}_{9}^{\left(2\right)}$	$s_{9}^{\left(1\right)},\;s_{9}^{\left(2\right)}$
10	+ − − +	${\underline{g}}_{10}^{\left(1\right)},\;{\underline{g}}_{10}^{\left(2\right)}$	${\bar{g}}_{10}^{\left(1\right)},\;{\bar{g}}_{10}^{\left(2\right)}$	$s_{10}^{\left(1\right)},\;s_{10}^{\left(2\right)}$
11	− + − +	${\underline{g}}_{11}^{\left(1\right)},\;{\underline{g}}_{3}^{\left(2\right)}$	${\bar{g}}_{11}^{\left(1\right)},\;{\bar{g}}_{11}^{\left(2\right)}$	$s_{11}^{\left(1\right)},\;s_{11}^{\left(2\right)}$
12	+ + − +	${\underline{g}}_{12}^{\left(1\right)},\;{\underline{g}}_{12}^{\left(2\right)}$	${\bar{g}}_{12}^{\left(12\right)},\;{\bar{g}}_{12}^{\left(2\right)}$	$s_{12}^{\left(1\right)},\;s_{12}^{\left(2\right)}$
13	− − + +	${\underline{g}}_{13}^{\left(1\right)},\;{\underline{g}}_{13}^{\left(2\right)}$	${\bar{g}}_{13}^{\left(1\right)},\;{\bar{g}}_{13}^{\left(2\right)}$	$s_{13}^{\left(1\right)},\;s_{13}^{\left(2\right)}$
14	+ − + +	${\underline{g}}_{14}^{\left(1\right)},\;{\underline{g}}_{14}^{\left(2\right)}$	${\bar{g}}_{14}^{\left(2\right)},\;{\bar{g}}_{14}^{\left(2\right)}$	$s_{14}^{\left(1\right)},\;s_{14}^{\left(2\right)}$
15	− + + +	${\underline{g}}_{15}^{\left(1\right)},\;{\underline{g}}_{15}^{\left(2\right)}$	${\bar{g}}_{15}^{\left(1\right)},\;{\bar{g}}_{15}^{\left(2\right)}$	$s_{15}^{\left(1\right)},\;s_{15}^{\left(2\right)}$
16	+ + + +	${\underline{g}}_{16}^{\left(1\right)},\;{\underline{g}}_{16}^{\left(2\right)}$	${\bar{g}}_{16}^{\left(1\right)},\;{\bar{g}}_{16}^{\left(2\right)}$	$s_{16}^{\left(1\right)},\;s_{16}^{\left(2\right)}$

**Table 2 bioengineering-12-00516-t002:** AI model 3, main effect: accuracy.

Term	Effect	Coef	T-Value	*p*-Value
$A_{3}$	0.006	0.003	2.09	0.041
$B_{3}$	0.009	0.004	2.92	0.005
$C_{3}$	−0.002	−0.001	−0.75	0.456
$D_{3}$	0.021	0.011	7.04	0.000

**Table 3 bioengineering-12-00516-t003:** Top Five Combinations for Model 3.

$A_{3}$	$B_{3}$	$C_{3}$	$D_{3}$	Acc.	Sen.	Spe.	Pre.	F1-S.
4	16	2	16	0.93	0.91	0.94	0.92	0.90
4	16	3	16	0.92	0.91	0.92	0.91	0.90
2	16	2	16	0.91	0.90	0.91	0.90	0.90
4	4	2	16	0.90	0.90	0.90	0.90	0.90
2	16	3	16	0.89	0.88	0.90	0.90	0.90

**Table 4 bioengineering-12-00516-t004:** Performance: three models.

Model	Acc. (s.e.)	Sen. (s.e.)	Spe. (s.e.)	Pre. (s.e.)	$F_{1}$-S (s.e.)
Model 1	0.88 (0.03)	0.88 (0.03)	0.89 (0.02)	0.88 (0.03)	0.87 (0.03)
Model 2	0.68 (0.05)	0.67 (0.06)	0.69 (0.05)	0.69 (0.05)	0.68 (0.05)
Concatenated	0.93	0.91	0.94	0.92	0.90
Model	(0.04)	(0.04)	(0.04)	(0.03)	(0.04)

## Data Availability

The raw data supporting the conclusions of this article will be made available by the authors on request.

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
