# Peer review of "Automated Early-Stage Glaucoma Detection Using a Robust Concatenated AI Model"

_bioengineering, 2025, doi:10.3390/bioengineering12050516_

Round 1
Reviewer 1 Report
Comments and Suggestions for Authors
This paper introduces a comprehensive AI model designed for the automated detection of early-stage glaucoma. It combines two types of input features: unstructured image data derived from fundus images and structured numerical features, including macular thickness and peri-papillary retinal nerve fiber layer (RNFL) thickness. The model utilizes a convolutional neural network (CNN) to process image features and an artificial neural network (ANN) to analyze numerical data. Hyper-parameters are optimized through a design of experiments (DOE) approach. The proposed model demonstrates impressive performance metrics, with accuracy, sensitivity, specificity, precision, and F1-score all-surpassing 0.90, thus outperforming current alternative models. The paper's topic is interesting and publishable; however, significant revisions are required before publication.
Comments for authors
Comment 1. The dataset used is from a single hospital in Taiwan. How to ensure the model's generalizability to diverse populations, especially given the potential variations in glaucoma presentation across different ethnic groups? Discuss this in the manuscript.
Comment 2. How does DOE compare to other optimization techniques like grid search or Bayesian optimization?
Comment 3. The comparison with state-of-the-art models, including those using deep learning and other AI techniques, is required to be discussed comprehensively.
Comment 4. Comprehensively assess the clinical limitations, potential toxicities, or lack of broad applicability.
Comment 5. The model's computational requirements are not discussed. How efficient is the model in terms of training and inference time, especially for real-time applications?
Comment 6. The conclusion section summarizes the contributions but does not discuss limitations or future perspectives. What are the limitations of the current model?
Comment 7. The present form of the manuscript contains several errors and typos that hinder clarity and may alter the intended meaning. I encourage authors to proofread and correct grammatical issues to enhance readability.
End of the report.
Comments on the Quality of English LanguageThe present form of the manuscript contains several errors and typos that hinder clarity and may alter the intended meaning. I encourage authors to proofread and correct grammatical issues to enhance readability.
Reviewer 2 Report
Comments and Suggestions for Authors
The research paper by Dr. Wheyming Songa and Dr. Ing-Chou Lai, entitled “Automated Early-Stage Glaucoma Detection Using a Robust Concatenated AI Model”, presents a novel artificial intelligence model for detecting early-stage glaucoma. This model was tested on 431 data points (204 healthy cases and 227 glaucoma cases, including 127 early-stage and 100 moderate or severe cases) provided by Kaohsiung Chang Gung (KCG) Memorial Hospital in Taiwan, demonstrating an accuracy of 0.93 for early-stage glaucoma detection.
The paper exhibits a robust structure, with the experimental design, methodological procedures, data collection, and data treatment based on the AI model thoroughly described. I believe that this work deserves to be published in Bioengineering without any changes to its scientific content. However, the paper’s format does not comply with the journal's editorial guidelines and should be revised accordingly. For example:
- ABSTRACT: Remove abbreviations that do not refer to frequently repeated terms. Use abbreviations appropriately within the main text.
- MAIN TEXT: it should be organized into sections and subsections (i.e., Introduction, Materials and methods, Results, Discussion)
- REFERENCES: References should be numbered sequentially as they appear in the main text. Revise all references accordingly.
Round 2
Reviewer 1 Report
Comments and Suggestions for Authors
I recommend accepting the paper in its present form.